# New Edible Packaging Material with Function in Shelf Life Extension: Applications for the Meat and Cheese Industries

**DOI:** 10.3390/foods9050562

**Published:** 2020-05-02

**Authors:** Roxana Gheorghita (Puscaselu), Sonia Amariei, Liliana Norocel, Gheorghe Gutt

**Affiliations:** 1Department of Human and Health Development, Stefan cel Mare University of Suceava, 720229 Suceava, Romania; liliana.norocel@usm.ro; 2Faculty of Food Engineering, Stefan cel Mare University of Suceava, 720229 Suceava, Romania; sonia@usm.ro (S.A.); g.gutt@fia.usv.ro (G.G.)

**Keywords:** bio-based polymers, food, preservation, time testing, zero-waste

## Abstract

Nowadays, biopolymer films have gained notoriety among the packaging materials. Some studies clearly test their effectiveness for certain periods of time, with applicability in the food industry. This research has been carried out in two directions. Firstly, the development and testing of the new edible material: general appearance, thickness, retraction ratio, color, transmittance, microstructure, roughness, and porosity, as well as mechanical and solubility tests. Secondly, testing of the packaged products—slices of cheese and prosciutto—in the new material and their maintenance at refrigeration conditions for 5 months; thus, the peroxide index, color, and water activity index were evaluated for the packaged products. The results emphasize that the packaging is a lipophilic one and does not allow wetting or any changes in the food moisture. The results indicate the stability of the parameters within three months and present the changes occurring within the fourth and fifth months. Microbiological tests indicated an initial microbial growth, both for cheese slices and ham slices. Time testing indicated a small increase in the total count number over the 5-month period: 23 cfu/g were found of fresh slices of prosciutto and 27 cfu/g in the case of the packaged ones; for slices of cheese, the total count of microorganisms indicated 7 cfu/g in the initial stage and 11 cfu/g after 5 months. The results indicate that the film did not facilitate the growth of the existing microorganisms, and highlight the need to purchase food from safe places, especially in the case of raw-dried products that have not undergone heat treatment, which may endanger the health of the consumer. The new material tested represents a promising substitute for commercial and unsustainable plastic packaging.

## 1. Introduction

The demand for food has increased steadily due to the modernization and growth of the population. Packaging is an important stage in the food industry, being responsible for maintaining the products’ quality and freshness, during transport, storage, and handling until consumption [1]. In general, the packaging materials used are multilayer types—complex, difficult to sort, and impossible to recycle—since they are based on plastic. The use of these packages has become unstoppable, precisely due to plastic’s ability to protect the packaged product. However, problems arise when packaging is obtained from non-renewable resources that consume and pollute the environment, thus endangering the health of humans and animals as well.

Several efforts have been made so as to extend the shelf life of food products and overcome the environmental problems of wasting packaging at the same time. The research has focused on the use of biopolymers. Researchers are studying combinations of polysaccharides with other materials to improve the barrier and mechanical properties in order to obtain biopolymers that could replace synthetic polymers [2]. Edible films and coatings are considered to be the biggest concern of the food industry today [3]. Biodegradable films and coatings and flexible and thin matrices for wrapping or coating are used in a diversity of product groups and industries [4]. In addition to their biodegradability, the use of edible films can represent an economical and feasible way to assure and preserve the quality, safety and nutritional value of products during the entire food chain [5]. Numerous studies have highlighted the similarity in terms of properties between biodegradable films and polymeric materials [6]. In contrast, biopolymer-based packaging is environmentally friendly, completely biodegradable or edible, obtained from renewable resources, and easily obtained at low cost. This packaging does not produce waste and can widely be applied to food packaging, fruit coatings, pharmaceuticals, and cosmetics [7]. Most of the present packaging is for single-use and disposed of after a first-use [8]. This category also includes sliced foods, such as sausages or cheeses. In general, these sliced foods come in complex packages, which usually contain successive layers of polyethylene/polypropylene, metal layers, and glue for bonding. The shelf-life of fresh meat is generally evaluated by monitoring the microbiological and sensorial changes of the product during storage time. In addition to these traditional methods, chemical metabolites result in microbiological deterioration of food products [9].

Nowadays, there is a consumer trend to use ready-to-eat foods, including pre-packed and pre-cooked products that come in different presentations. The transformation into ready-to-eat food involves additional steps, such as cutting, slicing, dicing, and packaging aimed at facilitating its consumption at home [10]. These can lead to contamination with microorganisms, especially pathogenic bacteria [11].

Seaweed-based polysaccharides are an interesting example of biopolymers. The films and coatings derived from such sources have good oxygen vapor barrier properties and are impervious to fats and oils [12].

Sodium alginate is a polysaccharide intensely used due to its special properties: biocompatibility, biodegradability, and ability to form good films [13]. Extracted from brown seaweed, it is intensively used in the food industry as a texturing and thickening agent. Annually, approximately 40,000 tons of alginate are used for commercial purposes in the food industry or other industries [14]. In general, alginate films have low water barriers [15] but adequate tensile strength, despite the negative charge of molecules [16]. To improve them, the industry has always looked for alternatives. One of these alterantives is mixing the sodium alginate with other hydrocolloids [17,18].

Agar is a polysaccharide extracted from red seaweed that is intensively used in packing film development due to its high mechanical strength and moderate solubility properties. It has synergism and has been tested in combination with other hydrocolloids, such as starch [19], carrageenan [20], protein [21], and gum [22].

Stevia was added to improve the film properties; this represents the ability to form clear, transparent, and glossy films, as noticed in a previous study [23]. Stevia leaves contain phenolic compounds with antimicrobial and antioxidant properties, vitamin C, carotenoids, and chlorophyll in high amounts, in addition to stevia glycosides [24].

Plasticizers–glycerol, in this case, are hydrophilic molecules such as polyols, which are supplementary to the film-forming materials, with the purpose of developing the physical and mechanical properties of the film [25].

The present study follows the development of a natural, completely edible packaging material based on biopolymers, used for packing meat products and sliced cheeses. Obtained entirely from biopolymers, the characteristics of the film promote it successfully for the substitution of the classic packaging materials, based on oil. In order to identify the capacity of use as packing material for foodstuffs, the material has been tested from physico-chemical and mechanical points of view. At the same time, solubility and food safety tests were performed, so that once obtained and tested, the film received applicability. Thus, fresh cheese and fresh prosciutto, an Italian specialty based on meat, were packaged and kept under refrigeration conditions for 5 months. After 3 months (the average storage period in the case of classic packaging), the products were tested in order to observe the freshness and the changes occurred. After this period, they were kept for two more months, in order to obtain new additional data regarding the possibility of extending the shelf life of the products, as a result of the used packaging.

## 2. Materials and Methods

### 2.1. Materials

All the substances used—agar, sodium alginate, glycerol, and stevia powder—were supplied from Sigma Aldrich Company. Prosciutto, an Italian dry-cured ham, was bought from a multinational store, and cheese from local producers. Throughout the testing period, the samples were kept in a refrigerator at the temperature of 4 °C. Compact dry culture media for microbiological analysis was purchased from a Romanian agent who intermediates the sale of such products from Japanese producers.

### 2.2. The Development of the Edible Film Used for Packaging

For the development of packing material with a surface of 30 × 60 cm, 1.95 g sodium alginate, 1 g agar, 0.005 g stevia, 1 g glycerol, and 150 mL water were used. The method of obtaining was performed according to those described by Puscaselu et al. [26]: the film-forming solution was obtained by homogenization at 450 rpm and kept for 30 min at 90 ± 2 °C, and then was poured onto a drying silicone support. The solution thus obtained was maintained at ambient temperature (23 ± 2 °C) until completely dry (approximately 47 h).

### 2.3. Testing of Physical Characteristics

After development, the film was tested in order to identify physical and optical characteristics. First of all, it was visually and olfactory appreciated. The thickness was measured using a digital micrometer with an accuracy of 0.001 mm (Mitutoyo, Kawasaki, Japan) and was established by reading in seven random areas of the film surface, and the value represents their arithmetic mean.

The retraction ratio is an important parameter for the industry. Thus, by establishing the thickness of the film and knowing the value of the retraction ratio, the manufacturer can establish from the beginning of the process the thickness of the film-forming solution, so that, finally, it has a material with well-established dimensions.

The retraction ratio is calculated according to the following formula [27]:(1)$$R\;\left(\%\right)=\frac{\mathrm{film}\;\mathrm{solution}\;\mathrm{thickness}-\mathrm{dry}\;\mathrm{film}\;\mathrm{thickness}}{\mathrm{film}\;\mathrm{solution}\;\mathrm{thickness}}\;*\;100$$

The results noted represent the sum of five determinations in different areas of the film surface.

The color was evaluated by the CieLAB method, using the Chroma Meter CR400 (Konika Minolta) and represents the sum of readings in at least seven areas of the film surface. The transmittance was tested using the Ocean Optics HR 4000 CG–UV–NIR spectrophotometer (Ocean Optics, 830 Douglas, AZ, USA) through repeated readings at 660 nm wavelength.

The microstructure of the film, as well as the roughness and porosity, were identified and analyzed using the MahrSURF CWM 100 confocal microscope. Mountain Map^®^ software (Version 7, Digital Surf, Lavoisier, France) was used to analyze and interpret microscopic images.

### 2.4. Solubility Testing

In order to be used as packaging material, the film was tested for moisture content (MC), given the composition it had. In this sense, the moisture was evaluated according to the method described by Puscaselu and Amariei [28]: 3 × 3 cm film samples were weighed, dried in the hot air oven at 110 °C for 24 h, and reweighed up to constant mass.

The results were recorded after applying Formula (2):(2)$$MC\;\left(\%\right)=\frac{W_{0}-W_{1}}{W_{1}}\;*\;100$$
where *W*_0_ represents the mass of the film before drying (g), and *W*_1_ the mass of the dried film (g).

Testing the solubility of the material in water (WS) involved the use of film samples with similar dimensions (3 × 3 cm). These were weighed and rehydrated by maintaining 8 h in a 50-mL vial of distilled water. After this time, they were removed from the liquid, gently buffered with filter paper in order to remove excess water and subjected to drying in the oven, at a temperature of 110 °C, for 24 hours. After this period, they were reweighed [29].
(3)$$WS\;\left(\%\right)=\frac{W_{0}-W_{1}}{W_{0}}\;\times \;100$$
where *W*_0_ represents the mass of the film before water immersion (g), and *W*_1_ the mass of the film after drying in the hot air oven (g).

The swelling ratio index is an important parameter that characterizes the film in terms of its stability after immersion and for a certain time (1–20 min) in water with a temperature of 23 ± 2 °C. The film samples, measuring 3 × 3 cm, are weighed before and after immersion. The results were noted after applying the following formula:(4)$$SR\;\left(\%\right)=\frac{W_{t}-W_{0}}{W_{0}}\;\times \;100$$
where *W_t_*—the mass of the film after immersion a moment *t* (g), and *W*_0_—the mass of the dry film (g).

The rehydration ratio (*Rr*) describes the ability of the material to rehydrate after drying. Therefore, for testing, films with a well-known size of 3 × 3 cm are weighed, dried, similar to the samples tested for water solubility, kept for 8 h in containers with 50 mL of distilled water, reheated for 24 h at 110 °C and then reweighed.
(5)$$Rr\;\left(\%\right)=\frac{W_{1}-W_{0}}{W_{0}}\;\times \;100$$

The results are noted according to Formula (5), where *W*_1_ represents the mass of the dry film before immersion (g), and *W*_0_ the mass of the dry film a second time in the oven (g).

The water activity index was measured using AquaLab equipment (ICT International, Amirdale NSW 2350, Australia); the measurements were made by five readings at 22.5 ± 2 °C.

### 2.5. Mechanical Properties

In order to determine the mechanical performance, the test samples were cut according to the standard STAS ASTM D882 (Standard Test Method for Tensile Properties of Thin Plastic Sheeting) [30] as follows: three film samples were cut to set a size of 100 × 10 mm. The determinations were performed in triplicate.

The tests were performed using the ESM 301-Mark 10 texturometer (Stefan cel Mare University, Suceava, Romania) using the gripping attachments for thin films and sheets (Addex Design, Sibiu, Romania). A loading cell of 5 kN was attached to the equipment, and the speed was set to 10 mm/min. The working temperature was 26.3 °C.

The tensile strength (*TS*) was calculated according to Formula (6). The elongation (*E*) is the capacity of the material to stretch until breaking, according to Formula (7), where *F*—the maximum force applied to the surface (*S*), and Δ*l* represents the difference between the initial length of the film (*l*) and the final length, mm.
(6)$$TS,\left(\mathrm{MPa}\right)=\frac{F}{S}$$
(7)$$TS,\left(\mathrm{MPa}\right)=\frac{F}{S}$$

### 2.6. Application of the Edible Films on Packaged Products

To determine the packing capacity of the newly obtained material, 5 g of cheese and 5 g of sliced prosciutto were packaged. Ten such samples were obtained for each product. The product was sealed with hot glue and kept under refrigeration conditions (temperature of 4 °C), protected from light and moisture, for 5 months. According to the indications on the commercial packaging, sliced products have a shelf life of about 3 months. Therefore, after three months, the packaged products were tested. For a more thorough verification and an evaluation of the capacity of the material used for packing, the products were kept under the same conditions for 5 months. Both at the initial moment, as well as after storage, the peroxide index, color, and water activity of the product were evaluated.

#### 2.6.1. Design of the Experiment

Fresh cheese and fresh smoked ham (prosciutto), an Italian specialty based on meat, were packaged using hot pressing and holding for 30 s at 170 degrees. Increasing the temperature to 190 degrees makes it impossible to manipulate the material due to the melting of the crystalline areas. The packaging used was a completely natural one, obtained from hydrocolloids, such as sodium alginate and agar, and plasticized with glycerol and water. Previous studies highlight the capacity of water molecules to improve the UV-barriers of the films, increasing film opacity [31], a desirable fact in the development of transparent packaging materials for the purpose presented above. In its composition was introduced powder of *Stevia rebaudiana*, due to its effect on improving the transparency of films.

The packaged products were kept under refrigeration conditions for 5 months. After 3 months (the average storage period in the case of a classic packaging), the products were tested in order to observe the freshness and the changes that have occurred. After this period, they were kept for another 2 months in order to obtain new additional data regarding the possibility of extending the shelf life of the products due to the packaging used.

#### 2.6.2. Determination of Peroxide Index

For lipid oxidation process analysis, the peroxide index was evaluated. Peroxide detection indicates rancidity in unsaturated fats and oils. Other methods are available, but the peroxide value is the most widely used. It indicates the extent to which an oil sample has undergone primary oxidation, while secondary oxidation may be determined through a p-anisidine test. The double bonds found in fats and oils play a role in autoxidation. Oils with a high degree of unsaturation are most susceptible to autoxidation. The best test for autoxidation (oxidative rancidity) targets peroxide values since peroxides are intermediates in the autoxidation reaction. Autoxidation is a free radical reaction involving oxygen that leads to deterioration of fats and oils, which form off-flavors and off-odors [32].

For determination, the fat from the research sample was extracted until completion with petroleum ether, a commonly used organic solvent, weighed and expressed as a percentage after removing the extraction solvent. The working method involves weighing 2–3 g of the sample. Total mass with extraction cartridge = wet sample mass.

In order to express the results: the wet sample was weighed and the result was given as gross fat/100 g with two decimal places [33]. Peroxide value (PV) is used most commonly as an indicator of the early stages of oxidation in fats and oils. Many chemical methods have been developed to quantify oxidative deterioration with the object of correlating data with off-flavor development [34].

#### 2.6.3. Determination of Color

The determination of color was made by the CieLAB method, with the use of the Chroma Meter CR 400 colorimeter (Konika Minolta, Japan), through repeated readings throughout the product surface.

#### 2.6.4. Water Activity Index Determination

To assess the water loss from sliced products, they were tested with AquaLab equipment (ICT International Amirdale, NSW 2350, Australia). The results noted represent the arithmetic mean of five readings.

#### 2.6.5. Microbiological Assessment

An extremely important test for food products, microbiological determinations were aimed at testing the total number of germs, *Escherichia coli*, *Staphylococcus aureus*, coliforms, enterobacteria, as well as yeasts and molds. For this purpose, Compact Dry-type plates with lyophilized culture media were used. In order to obtain information regarding microbial growth, the plates were kept in hot air oven according to the well-known time and temperature indications: 37 °C for 72 h for yeasts and molds and 48 h for the other microorganisms. The principle of the method involved the weighing of a 1-g sample and the addition of 9 mL of sterilized water. Then, 1 mL of the solution thus obtained was used to hydrate the culture media. Three such dilutions were performed.

### 2.7. Statistical Evaluation

The data were analyzed by principal component analysis (PCA) with XLSTAT software (2020, trial version, Addinsoft, NY, USA), which evaluated the correlations between the thickness of the film and texture parameters analyzed the variation and extracted the main components. Principal component analysis aims at the thickness (lower and higher) as active variables, and elongation, transmittance, water-solubility, moisture, tensile strength, adhesiveness, retraction ratio, and roughness as active observations. It covers 100% of the variability.

## 3. Results and Discussion

### 3.1. Evaluation of the Characteristics of the New Material

The film, noted as S1, presented a high gloss and transparency, was fine and pleasant to the touch, without visible asperities, pores, or cracks on the surface. It was without smell, with a sweet taste that was barely noticeable. It had good mechanical performances (breaking resistance and superior elasticity) and high solubility. It is perfect for packing powdery products, for wrapping fresh fruits and vegetables, sweets, and pastries, and for foils for the pharmaceutical industry. It presented with pores in the structure, with diameters between 6.72–12.50 nm and depths of 35.60–421.00 nm. With the exception of the pores, the surface of the film was very homogeneous, with no asperities or insoluble particles. There were no visible cracks on its surface (Figure 1).

The high value of the transmittance promotes the use of film packing for food products, as market studies have indicated the buyer’s desire to see the product through the packaging. The products with transparent packaging sell better than those packed in opaque materials. Moreover, such a high value of the transparency of the material will not affect the real color of the product. The values obtained after testing the material based on biopolymers are noted in Table 1.

Thickness is reduced, if we take into account the films obtained from agar and sodium alginate in other studies (73 µm in the study of Atef et al. [35], or 76 µm in the study of Siah et al. [36]).

There is a reduction of their mass, an aspect highlighted by the data in Table 2. We can conclude that the transfer was made from the inside to the outside, taking into account the weight reduction of the sample, as shown in Table 2. This aspect emphasizes that the packaging is a lipophilic one, thus not allowing wetting or any changes in the food moisture.

Principal component analysis (PCA) was performed in order to emphasize the difference in film thickness, and the results are presented in the biplot from Figure 2. The PCA method limited all data into two main components covering 100% of the variability (PC1—99.81% and PC2—0.19%). Analyzed parameters such as elongation, transmittance, water-solubility, moisture content, and elongation were grouped around the lowest thickness value of the film. Retraction ratio, adhesivity, and roughness are more representative of the films with a high thickness. Tensile strength had almost medium values and is located in the center of the biplot; this analysis does not present it as a principal component that is influenced by the film thickness.

The new film can substitute conventional packaging in terms of mechanical performance. The data are noted in Table 1.

The film is resilient and elastic. Its high resistance and high value of the breaking point strengthen the possibility of using it as a substitute for conventional materials. The adhesiveness of the material is given by the glycerol content and the addition of stevia. Regarding the moisture content of the material, it was 12.65% ± 0.120%. The solubility in water had values of 39.82% ± 0.033% and water activity index was 0.382 ± 0.033 (Table 1). Both the swelling ratio and the rehydration ratio increased directly proportionally with the immersion time, which is normal fact. Unlike other films made from the same polysaccharides, but with different amounts, it has not completely solubilized [6]. The values are in favor of using the film for packing sliced meat products, which usually have higher humidity than those that are marketed whole.

### 3.2. Evaluation of Packaged Products

The packaged products were sliced cheese and sliced prosciutto. Each 5-g of each product was packed in the obtained film. Ten such samples of 5-g each were tested during this study.

Figure 3 shows the way in which the sliced products were kept. In the case of meat slices, changes in fat color can be observed. Product degradation is correlated with the data obtained from the determination of the peroxide index. Peroxide value, the concentration of peroxide in an oil or fat, is useful for assessing the extent to which spoilage has advanced. However, the data obtained do not indicate the impossibility of consumption since the values are still low. The images show the stability of the packaging film. Even at handling, it did not disintegrate by breaking or tearing, maintaining its superior mechanical properties.

The evolution of the analyzed parameters of sliced cheese and prosciutto is presented in Figure 4 and Figure 5. As can be seen in Figure 4, the values of the analyzed parameters begin to change significantly after 90 days after packing. The biggest changes were the case of A * and L * color parameters, and water content.

The results of the color parameters indicate small changes, brightness being the most affected. In the case of the other parameters analyzed (w_a_—water activity index, peroxide index), which can characterize the freshness and lipid oxidation of the product, the most observable changes appear to the peroxide index after 50 days from packaging.

The results may also be due to the porosity of the film since it has not disintegrated. The pores present at its surface allowed the transfer of moisture with the external environment. This indicates the need to improve this material, either by modifying the composition (higher glycerol content) or by adding other biopolymers, such as those protein-based (which have a higher vapors or gases stability).

Regarding the microbiological tests, 23 cfu/g were found of fresh slices of prosciutto and 27 cfu/g in the case of the packaged ones after 5 months storage under refrigeration conditions. For slices of cheese, however, the total count of microorganisms indicated 7 cfu/g in the initial stage and 11 cfu/g after 5 months. The results indicate that the film did not facilitate the growth of the existing microorganisms, taking into account the small differences during this time. However, we are worried about the initial micobiological growth, especially in the case of prosciutto slices. Being a raw-dry food, the possibility of growth and proliferation of microorganisms is higher. That is why it is important to buy food from well-known stores or, when buying from local producers, their source must be known. The other microorganisms, *Escherichia coli*, *Staphylococcus aureus*, enterobacteriaceae, coliforms, yeasts or molds, did not grow on the culture media. A similar situation was also reported by the Viuda-Martos et al. [37]. They argue that the absence of yeasts and molds is due to the aseptic process of slicing and packaging, together with the presence of sodium chloride in the packaged products. Other results, presented by Ouatarra et al. [38], reported that indigenous *Enterobacteriaceae* in meat products (bologna, beef pastrami, and cooked ham) were inhibited by the use of antimicrobial films containing chitosanand acetic acid. Similarly, various antimicrobial substances can also be included in these films.

## 4. Conclusions

The present study aims at obtaining a biopolymer packing material, its testing and characterization, as well as its applications in coating slices of ham and cheese. Tested over a period of 5 months, the results indicate that the packaging is able to successfully replace the conventional, multilayer-type packaging that is so harmful to the environment. Packaged in conventional foils, sliced products have a shelf life of up to 3 months. The fact that these products have been tested over a period of 5 months indicates the film’s ability to maintain the product characteristics.

The solution presented is not an expensive one. The film surface used to pack a product (such as cheese or ham slices) costs about 1.15 euros per package. These costs represent only the price of the substances, not the utilities or labor force required. Certainly, the price can be considerably reduced when the material is obtained at industrial level. It is also necessary to point out that the producing companies do not have to pay the extremely expensive waste taxes, given that the product generates zero waste. Moreover, to save the used material, the products can be packaged in other shapes or sizes, as the films are much more flexible than the ordinary trays used as a support for slices of ham, for example.

However, it is intended to improve the performance of this material, in order to reduce the porosity and eliminate any possible transfers with the external environment. Thus, future studies will focus on this aspect.

## Figures and Tables

**Figure 1 foods-09-00562-f001:**
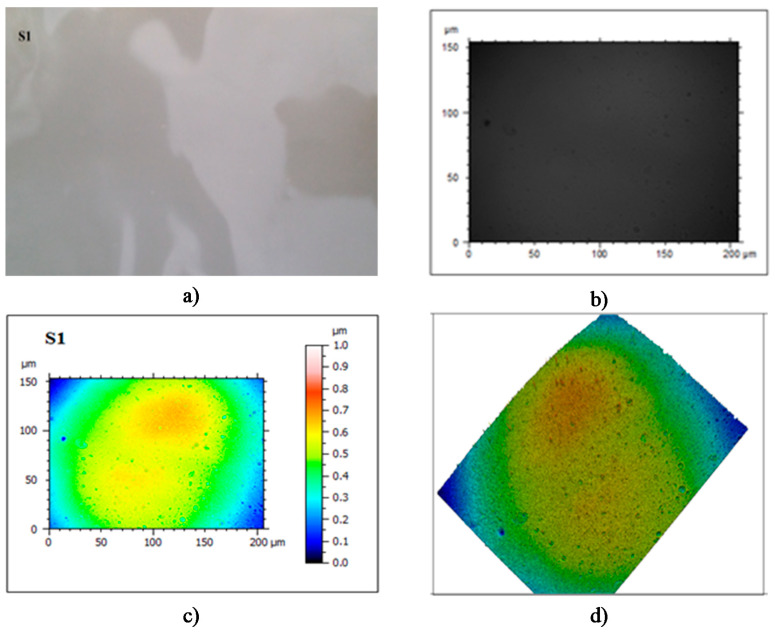
Images of the film. (**a**) Real image; (**b**–**d**)—images captured using the confocal microscope.

**Figure 2 foods-09-00562-f002:**
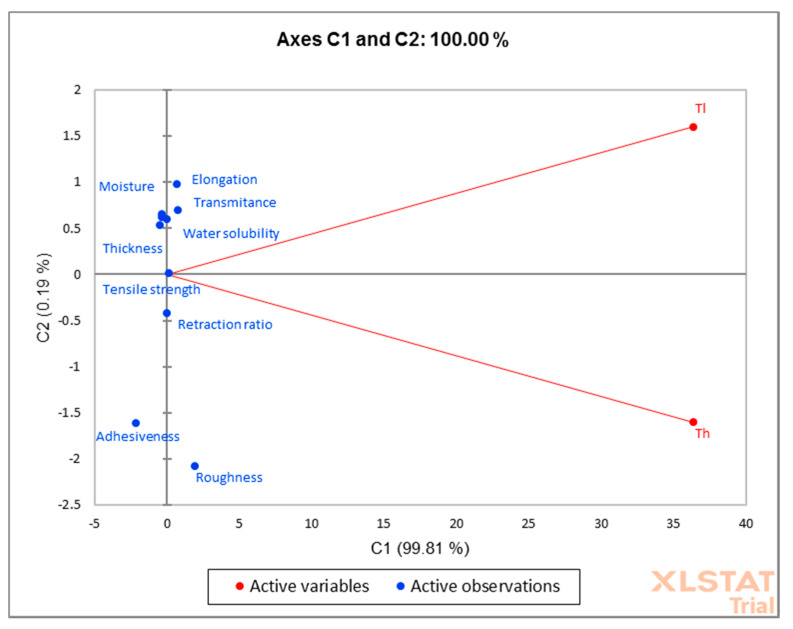
Principal component analysis (PCA) of the dataset consisting of analyzed parameters of film (where Th represent the highest thickness of the film and the Tl—the lowest value of film thickness).

**Figure 3 foods-09-00562-f003:**
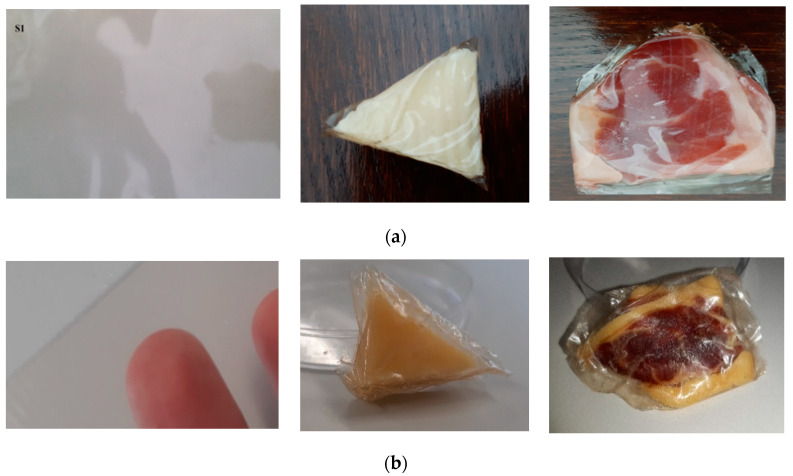
The film and the products packed at the initial moment (**a**) and after 5 months of maintenance in refrigeration conditions (**b**).

**Figure 4 foods-09-00562-f004:**
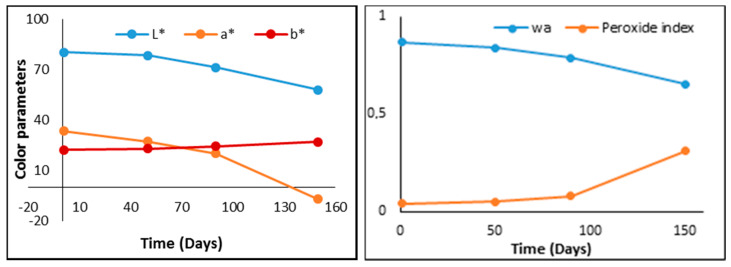
Evolution in time of analyzed parameters of sliced cheese (peroxide index is expressed in meq O2/kg of samples).

**Figure 5 foods-09-00562-f005:**
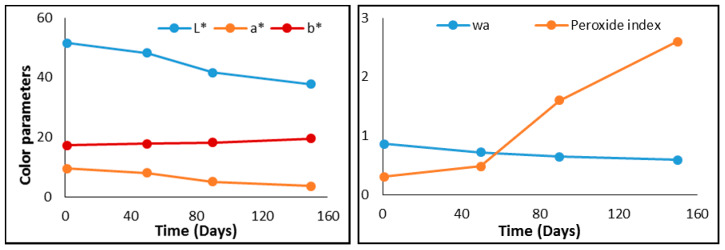
Evolution in time of analyzed parameters of prosciutto (peroxide index is expressed in meq O2/kg of samples).

**Table 1 foods-09-00562-t001:** Characteristics of the new material tested.

Physical and Optical Properties							
Thickness, µm	Retraction Ratio, %		Transmittance, %	Color			
				L *		A *	B *
47.20 ± 0.18	35.78 ± 0.13		92.19 ± 0.88	93.79 ± 0.24		−6.79 ± 0.04	20.03 ± 0.57
**Mechanical Properties**							
**Tensile Strength, MPa**	**Elongation, %**		**Breaking Point, g**			**Roughness, nm**	**Adhesiveness, g * s**
3.00 ± 0.34	84.80 ± 0.81		6646.995 ± 3.563			175.00 ± 0.8	−116.11 ± 0.98
**The Rehydration Capacity**							
**Time (min)**	**1**	**3**	**5**	**7**	**10**	**15**	**20**
*Rr*, %	2394	2770	2840	3676	4135	4570	5022
*SR*, %	665,000	683.840	1,118,000	1,267,800	1,451,900	1,760,740	2,087,020

Note * *Rr*—rehydration ratio, *SR*—swelling ratio.

**Table 2 foods-09-00562-t002:** Changes in the mass of packaged products during the testing period.

	Initial Moment (t_0_)	Month 3	Month 5
Sliced and packaged cheese, g	5.06 ± 0.005	4.89 ± 0.25	4.08 ± 0.46
Sliced and packaged prosciutto, g	5.09 ±0.004	4.75 ± 0.12	4.17 ± 0.03

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
