# Peer review of "New Edible Packaging Material with Function in Shelf Life Extension: Applications for the Meat and Cheese Industries"

_foods, 2020, doi:10.3390/foods9050562_

Round 1

Reviewer 1 Report

The development of civilization and the increase in the consumption of all kinds of goods has resulted in an overproduction of packaging, which is mostly non-biodegradable and pollutes the environment. Unfortunately, this is a global problem. In this context, any proposal related to the use of innovative packaging is extremely valuable. The authors undertook very interesting research not only from both scientific and implementation point of view. The results of the research are transparently presented to the reader, not raising objections to the achievement of the research goal set by the authors. They were illustrated both numerically in tables and diagrams using appropriate statistical methods and pictures of the research material in different phases of the research. However, I would suggest to the authors that:

· the Abstract should indicate specifically the research methods used. Adding this information would certainly be an element encouraging other researchers working in this field to refer to this article.

· literature studies and Introduction should be extended. The quality of scientific reports and their topicality is acceptable, but the quantity (23 items) needs to be corrected (I suggest to extend it),

· the article should be restructured: this suggestion concerns the Chapter 2. MATERIALS AND METHODS. Breaking down the section into additional subsections containing one sentence deteriorates readability. Please merge them if possible or provide a more detailed characterization, for example for the materials used (2.1. Materials);

· in the conclusions (4. CONCLUSIONS), please provide an approximate cost of the packaging they propose (if possible and if the authors have an idea about its level).

I hope that my suggestions for changes will encourage both scientists and representatives of food enterprises to read the article and cooperate with the Authors.

Author Response

Dear Ms / Mr Reviewer,

Thank You for your feedback.

We made the suggested changes and I hope that in the current form, the manuscript presents the information in a much more accurate and easy to understand form. 

Best Regards,

Dr. Roxana Gheorghita

Reviewer 2 Report

Revision

New edible packaging material with function in shelf life extension. Applications for the meat and cheese industries

The manuscript describes the results of the study on the using biopolymers contained edible components: agar, sodium alginate, glycerol and stevia powder. Although the topic is up of date and valuable the presented form manuscript needs deep revision.

Major comments:

  1. The introduction should be rearranged and revised. The authors are encouraged to phrase a hypothesis and to state specific major novel contributions reported in their manuscript. Lines 40-51 present summary of the study. It is not appropriate. In introduction Authors should present the current state of knowledge and end-defined hypothesis. Also introduction should not describe commonly well-known knowledge. Only some importance latest facts.
  2. Materials and Methods should be supplemented with diagram describing obtaining the edible films.
  3. There is a lot of mistakes which must be eliminated:

For example, The Authors say “For the testing of freshness, the peroxide and iodine index were evaluated”. This not true, peroxide value is used for analysis lipid oxidation process. Iodine value for analysis/measure unsaturation degree.

  1. Please delete the results about iodine value as I mentioned above this method is not adequate for analysis changes in fat extracted from products during storage.
  2. Description presented in: 2.3.1. Determination of peroxide and iodine indices (Lines 156-177), is unacceptable. Please revise it describe in brief lipid extraction process and used peroxide value procedure (with proper citations).
  3. In part of Methods Authors should separate methods for study packing films/materials and method for analysis cheese and ham during storage.
  4. Line 194, description  of statistical analysis is not sufficient.
  5. Authors proposed three small table (1-3) presenting results of properties of film. Please combine into one Table.
  6. This is true that Authors analysed 5 g food samples ? Why ?  This is unreal.
  7. I strongly suggest correction in grammar and style of English language.
  8. Conclusions need revision: Please not use phrases like “ the results obtained are good”, “Normally,”
  9. Revising part of Results and Discussion authors should focused on the results of microbiological quality of products sored in developed films.

 Minor comments:

  1. There is many mistakes in the text:

For example in the Abstract:

“prosciutto” this is Italian.

“microbiological load” what does it means ?

  1. Also there is a lot of mistakes in figures.

Author Response

Dear Ms/Mr Reviewer,

Thank you for your comments and your feedback.

We made the suggested changes and I hope that in the current form, the manuscript presents the information in a much more accurate and easy to understand form.

As for the translation of the "prosciutto" term, it refers to an Italian dry-cured ham, according to Wikipedia. We did not find information that this would be presented under another name, that's why we didn't change it. However, we accept the modification of this term in case another form of expression exist or will be communicated to us. 

Best regards,

Dr. Roxana Gheorghita

Round 2

Reviewer 2 Report

Revision 2: New edible packaging material with function in shelf life extension. Applications for the meat and cheese industries:

I found that manuscript was much improved. However,  there are some mistakes in the manuscript which in my opinion should be eliminated to obtain final accepted version of article. Manuscript needs serious English language revision.

Dear Authors, In Introduction part I started correct manuscript and arrange text.

Comments:

Abstract:

Lines 15 – 20: The part describing results needs revision.

 Introduction:

  1. Lines 33-44: should be supplemented with current state of knowledge about biodegradable materials which can be used for food packing. I suggest Authors to familiarize with some published papers eg. Polysaccharide-based films and coatings for food packaging: A review Food Hydrocolloids Vol. 68, July 2017, Pages 136-148
  2. Authors are asked to improve/revise the aim of the study:

The present study followed the development of a natural, completely edible packaging material, based on biopolymers, used for packing meat products and sliced cheeses.

  1. Other part of the text I suggest to move in the MATERIALS AND METHODS

2.1. Design of the experiment

Fresh cheese and fresh smoked ham (prosciutto), an Italian specialty based on meat, were packaged with using (Authors Please supply…………….). The packaging used was a completely natural one, obtained from hydrocolloids such as sodium alginate, agar, plasticized with glycerol and water. In its composition was introduced powder of Stevia rebaudiana, due to its effect on improving the transparency of films..

The packaged products were kept under refrigeration conditions for 5 months. After 3 months (the average storage period in the case of a classic packaging) the products were tested in order to observe the freshness and the changes occurred. After this period, they were kept for another 2 months, in order to obtain new additional data regarding the possibility of extending the shelf life of the products, due to the packaging used.

2.2. Materials

2.3. Methods

Page 5:

  1. I suggest to remove description and results regarding iodine value. Iodine value is used for analysis/measure unsaturation degree it is not reflect to changes in food lipids during storage.
  2. Lines: 346-351. Must be improved (Extraction of lipid from food (and other sources) is always provided with using organic solvents e.g hexan. Authors should write one sentence about using solvent/s.
  3. Lines 351 – 354: there is no need describing other methods then Peroxide Value. Please delete unnecessary sentences.
  4. Line 355: Please delete “properties”.
  5. Line 372: “The resultant data” is not correct, Please revise.

Results and discussion

  1. Lines 450-451: The sentence change into: There were no visible cracks on its surface (Figure 1).
  2. Line 453: Please delete the sentence “The values obtained after testing the material based on biopolymers are noted in table 1.” A reference to the table 1 insert at the line 458.

Author Response

Dear Reviewer,

Thank you for your comments and suggestions. 

We hope that this form of manuscript will meet your requirements.

Dr. Roxana Gheorghita - corresponding author